# Distribution of Polysulfide in Human Biological Fluids and Their Association with Amylase and Sperm Activities

**DOI:** 10.3390/molecules24091689

**Published:** 2019-04-30

**Authors:** Mayumi Ikeda, Yu Ishima, Victor T. G. Chuang, Maki Sakai, Hiroki Osafune, Hidenori Ando, Taro Shimizu, Keiichiro Okuhira, Hiroshi Watanabe, Toru Maruyama, Masaki Otagiri, Takaaki Akaike, Tatsuhiro Ishida

**Affiliations:** 1Department of Pharmacokinetics and Biopharmaceutics, Institute of Biomedical Sciences, Tokushima University, 1-78-1, Sho-machi, Tokushima 770-8505, Japan; c401741005@tokushima-u.ac.jp (M.I.); c401503044@tokushima-u.ac.jp (M.S.); c401503030@tokushima-u.ac.jp (H.O.); h.ando@tokushima-u.ac.jp (H.A.); shimizu.tarou@tokushima-u.ac.jp (T.S.); okuhira@tokushima-u.ac.jp (K.O.); ishida@tokushima-u.ac.jp (T.I.); 2School of Pharmacy, Monash University Malaysia, Jalan Lagoon Selatan, Bandar Sunway, Subang Jaya Selangor 47500, Malaysia; victor.chuang@monash.edu; 3Department of Biopharmaceutics, Graduate School of Pharmaceutical Sciences, Kumamoto University, 5-1 Oe-honmachi, Kumamoto 862-0973, Japan; hnabe@kumamoto-u.ac.jp (H.W.); tomaru@gpo.kumamoto-u.ac.jp (T.M.); 4Faculty of Pharmaceutical Sciences, Sojo University, 4-22-1 Ikeda, Kumamoto 860-0082, Japan; otagirim@ph.sojo-u.ac.jp; 5Department of Environmental Health Sciences and Molecular Toxicology, Tohoku University Graduate School of Medicine, Sendai 980-8575, Japan; takaike@med.tohoku.ac.jp

**Keywords:** polysulfide, biological fluids, circadian rhythm, aging

## Abstract

Intracellular polysulfide could regulate the redox balance via its anti-oxidant activity. However, the existence of polysulfide in biological fluids still remains unknown. Recently, we developed a quantitative analytical method for polysulfide and discovered that polysulfide exists in plasma and responds to oxidative stress. In this study, we confirmed the presence of polysulfide in other biological fluids, such as semen and nasal discharge. The levels of polysulfide in these biological fluids from healthy volunteers (*n* = 9) with identical characteristics were compared. Additionally, the circadian rhythm of plasma polysulfide was also investigated. The polysulfide levels detected from nasal discharge and seminal fluid were approximately 400 and 600 μM, respectively. No correlation could be found between plasma polysulfide and the polysulfide levels of tear, saliva, and nasal discharge. On the other hand, seminal polysulfide was positively correlated with plasma polysulfide, and almost all polysulfide contained in semen was found in seminal fluid. Intriguingly, saliva and seminal polysulfide strongly correlated with salivary amylase and sperm activities, respectively. These results provide a foundation for scientific breakthroughs in various research areas like infertility and the digestive system process.

## 1. Introduction

Thiols are one of the most important targets of posttranslational modification via redox reaction due to its nucleophilicity. When thiol is exposed to reactive oxygen species (ROS) or reactive nitrogen species (RNS), it is oxidized to SO_x_H or SNO [1,2]. Reversible modification of thiols forms disulfide bonds. Some electrophilic compounds, such as reactive aldehyde, can be modified to thiol irreversibly [3,4]. These modifications sometimes change protein structure and/or activities, thus thiols could be one of the important modulators of protein functions [5]. Recently, the investigation of reactive sulfur species (RSS) provided a paradigm shift in the redox biology of thiols. RSS is defined as sulfur-abundant molecules, such as cysteine persulfide (CysSSH) and glutathione persulfide (GSSH) [6,7]. These sulfur-bound sulfur atoms are called “sulfane sulfur” [8], which renders a stronger nucleophilicity effect to thiol by α effect [9]. For example, pKa of CysSSH is 4.3, while that of cysteine (CysSH) is 8.4 [9]. Thus, hydropolysulfide is a better target for posttranslational modification than thiol. Cysteine polysulfide is commonly measured by alkylation and reduction methods [10,11]. In this method, polysulfide and thiol are capped by weak alkylation agents like iodoacetamide, then treated with reductants including dithiothreitol (DTT) or 2-mercapto ethanol (2-ME). Thiol will not be reduced after alkylation, whereas polysulfide will be reduced by the reductants. 

The levels of low molecular RSS including CysSSH, GSSH, and CysSSSCys were detected in blood, heart, liver, brain, and lung [6]. Akaike et al. reported that CysSSH bound to tRNA preferentially rather than CysSH [12]. They also proved that cysteine polysulfides (CysSS_n_H) account for about 70% of total cysteine in protein during translation [12]. A good proportion of CysSS_n_H remains while the CysSS_n_H incorporated to protein is reduced by thioredoxin (Trx)/Trx reductase (TrxR) systems. These observations mean that CysSS_n_H is a natural component of proteins. Consequently, measuring the whole amount of polysulfides is therefore important for assessing redox balance.

On the other hand, little is known about the existence of polysulfide in secretory extracellular proteins. One of the reasons is that oxidative environment of fluids converts reduced form (CysSS_n_H) to oxidized form (CysSS_n_Cys) of polysulfides that cannot be detected by the alkylation method. Serum albumin constitutes approximately 60% of the serum proteins, hence being the most abundant protein in plasma. Many mammalian serum albumins have 35 residues of cysteine and only one of them exists in reduced form. Using alkylation and reduction method, P. Nagy et al. demonstrated that serum albumin acquired the reduced form of polysulfide after being treated with sodium hydrogen sulfide [11]. 

In a previous study, we have successfully developed a novel analytical method for quantifying the oxidized form of polysulfide, named as elimination method of sulfide from polysulfide (EMSP) [13]. This EMSP assay enables measurement of the polysulfide in biological fluids like plasma. The assay also revealed that human serum albumin (HSA) carries the oxidized form of polysulfide. Some of these biological fluids contain proteins that are common to plasma [14,15,16], so it is possible for them to also contain polysulfide, potentially demonstrating positive correlation with plasma polysulfide levels.

In this study, we examined the polysulfide content of biological fluids such as semen and nasal discharge in healthy subjects. By collecting various biological fluid samples from the same subjects, polysulfide levels in the biological fluids could be compared to that in plasma. We also examined the circadian rhythm of plasma polysulfide.

## 2. Results and Discussions

### 2.1. Determination of Polysulfide Level in Biological Fluids

We previously demonstrated that polysulfide in plasma exists mostly in HSA [13]. HSA is present not only in plasma but also in several biological fluids. In this study, polysulfide levels of plasma, tears, saliva, nasal discharge, and semen were measured. 

These biological fluids were collected from 9 healthy subjects, including 5 males. Semen was collected from 4 of the 5 male subjects, because only four informed consents for semen collection were obtained. The subject characteristics are summarized in Table 1. The average age was 28.44 years, and the average body mass index (BMI) was 20.85. The polysulfide level in plasma, tears, saliva, nasal discharge, and semen was determined using the EMSP method. As previously reported, the polysulfide level was about 7.5 mM for plasma, about 1 mM for tears, and about 41 μM for saliva [13]. The polysulfide of nasal discharge and seminal fluid was quantitated for the first time and it was about 400 μM and 600 μM, respectively. The correlation between plasma polysulfide levels and these biological fluids was examined. Results showed that there was no correlation between plasma and tears, saliva, or nasal discharge (Figure 1a–c). Interestingly, only semen showed a positive correlation with plasma (Figure 1d). The protein content of each biological fluid was examined to shed light on the correlation results.

### 2.2. Analysis of Protein Content of Biological Fluids

Because the component of tear fluid is similar to plasma, autologous plasma eye drops are widely used for the treatment of dry eyes [17]. However, a comparison of the high speed Triple TOF system analysis showed that the proportion of main proteins was different between plasma and tear fluids [18]. The most abundant proteins in tear fluids are lysozyme, lactoferrin, and lipocalin 1, whereas those of blood plasma are albumin and immunoglobulins. No correlation between the polysulfide in tear fluids and the plasma polysulfide level could be determined, due to the difference in protein compositions.

Saliva is an ideal specimen for clinical diagnosis because of the noninvasive sampling method. Proteomics analysis using two-dimensional gel electrophoresis (2D gel) revealed that some saliva proteins are also present in plasma [15]. This evidence indicates that those proteins may have transferred from plasma to saliva, while the other saliva proteins are produced locally by the salivary gland. Serum albumin is one of the abundant proteins in saliva besides salivary α-amylase. Another plasma protein, prolactin, is also found in saliva. These proteins are reported to come from gingival crevicular fluid (GCF). The protein composition of GCF is almost similar to that of plasma [19]. Despite this similarity in protein content, there is no correlation between saliva and plasma (Figure 1b), which may be caused by the ordinal oxidative stress in oral fluid [20].

Nasal discharge constituted of interstitial fluid, plasma, mucus, and nasal secretion [16]. Polysulfide levels, as well as protein levels, in one of the healthy subjects were 3–10 times higher than others, however, the polysulfide/protein molar ratio in nasal discharge did not correlate with plasma. This result suggests that the polysulfide in nasal discharge may be from other sources except plasma. 

The positive correlation of polysulfide levels between semen and plasma suggests that the protein composition of seminal fluid may be similar to that of plasma. Human seminal fluid is a secretion from the seminal vesicle, epididymis, prostate, and the urethral gland. The seminal fluid accounts for 95% of total semen [21]. Previous studies demonstrated that HSA constituted approximately 17.7% to 22.7% of the total protein in semen, while that of plasma is about 64% [14]. In contrast, immunoglobulins (alpha, beta, and gamma) occupy a higher ratio of the protein content in semen [14,22]. 

In addition to protein content, the difference of redox environment among these fluids may also contribute to the polysulfide level in each biological fluid examined. Oral environment is exposed to ROS produced by oral bacteria [23]. The eye is also exposed to ROS caused by wearing contact lens [24,25] or inflammations [26]. Compared to those two environments, ROS levels in seminal vesicle or testis might be very low in a healthy subject. Therefore, polysulfide in semen reflected the oxidative stress of plasma.

### 2.3. Effect of Age, Gender Difference, and BMI on Polysulfide Levels in Biological Fluids

Relationships among age, gender difference, BMI, and plasma polysulfide level were assessed. There is no statistically significant difference (*p* = 0.052), however, Figure 2a showed that plasma polysulfide tends to be higher with increasing age within the range investigated (22–43 years old). Aging is known as one of the risk factors of oxidative stress [27]. Therefore, we predicted that aging would decrease the amount of polysulfide, but the results indicated otherwise. This may be due to the age range examined in this study being rather narrow. Previous reports showed that aging decreases the ratio of the antioxidant glutathione/glutathione disulfide (GSH/GSSG) for the age range of 40 to 90 [28]. For the age range of below 40, the ratio was getting higher with increasing age. Further studies are required to understand the overall relationship between aging and polysulfide levels. On the other hand, there was no association between plasma polysulfide with gender difference or BMI (Figure 2b,c). 

Amylase activity was measured as previously described [29] (Figure 2d). Interestingly, it was shown that as the polysulfide level in the saliva increases, the amylase activity increases as well. Physical or psychosocial stress is known to increase the activity of amylase [30,31]. JL Kroll et al. reported that the level of hydrogen sulfide (H_2_S) in saliva increases with psychological stress [32]. Thus, polysulfide levels might associate with salivary amylase activity. Intriguingly, oxidation of cysteine residue on bacterial α-amylase is known to decrease its activity [33]. Further investigation is required to investigate whether polysulfide controls the activity of α-amylase.

### 2.4. Relationship Between Sperm Activity and Polysulfide Levels in Semen 

The polysulfide levels in semen showed a strong positive correlation with the amount of alive sperm measured by WST-8 (Figure 3a). Conversely, there was no correlation between polysulfide level and the semen volume or age (Figure 3b,c). Semen was centrifuged to separate seminal fluid from sperm, so that the polysulfide levels in seminal fluid and sperm could be determined. The results showed that most polysulfide was contained in seminal fluid (Figure 3d). These data suggested that the correlation between polysulfide levels in seminal fluid associated with sperm activity (Figure 3a) was most likely due to the redox activity of polysulfide. Several studies have reported that ROS damages sperm DNA and decreases sperm motility [34]. The presence of cysteine or glutathione improved the motility by suppressing ROS [35]. Hydrogen sulfide (H_2_S) also has been reported to prevent sperm from oxidative stress [36]. The present study on healthy volunteers showed that the ROS level in semen did not change by age, however, ROS of infertile men (>40 years) was significantly higher than that of men under 40 years of age [37]. Further study of the effect of semen polysulfide on these age-related ROS levels should lead to development of effective diagnostic tools of infertility. 

### 2.5. The Circadian Rhythm of Polysulfide Level in Plasma

Plasma hydrogen sulfide (H_2_S) concentration has been reported to exhibit diurnal fluctuations [38]. To investigate if the timing of sampling has any effect on the polysulfide level, we examined the circadian rhythm of plasma polysulfide levels. This is the first report to investigate circadian rhythm of plasma polysulfide. The plasma polysulfide level, measured by EMSP, tended to increase slightly from 12:30 pm to 21:30 and tended to decrease again by noon, but there was no statistically significant difference (Figure 4a). In addition, we also measured the polysulfide level in plasma using SSP4, which is a fluorescent probe of polysulfide. Figure 4b showed that the mean fluorescence intensity (MFI) significantly increased until 15:30 and fell by 3:30 at night. The polysulfide level at the time of 15:30 was significantly higher than that of 0:30 and 3:30. Next, the antioxidant activity of plasma was evaluated using the AAPH radical elimination method. The antioxidant activity increased around 15:30 (Figure 4c), but the plasma thiol level fell slightly around midnight (Figure 4d). The eliminated radical level at 15:30 was significantly higher than at 9:30 (Figure 4c).

The correlation of each parameter is shown in Figure 4e. The fluorescence intensity of SSP 4 and the radical scavenging activity of AAPH showed a positive correlation. Polysulfide level measured by EMSP increased from 12:30, reaching a maximum at 21:30. On the other hand, the SSP4 intensity decreased between 15:30 to 3:30 followed by an increase between 12:30 to 15:30. The discrepancy between EMSP and SSP4 results might be due to the reactivity of each reagent. SSP4 would attack cysteine residues on the protein surface only due to steric hindrance, whereas EMSP could react with polysulfide at all locations in a protein. In fact, the activity of AAPH radical elimination may have a similar rhythm to polysulfide measured by SSP4 because polysulfide on the surface may scavenge ROS easier than intramolecular polysulfide (Figure 4b–e). It is reported that H_2_S binds to HSA expeditiously, however, H_2_S levels did not affect plasma polysulfide level in this study. A previous report has shown that the plasma H_2_S level of mice at 7:00 is lower than 19:00 via 3-mercaptopyruvate sulfurtransferase activity [37]. Mice are nocturnal animals, thus plasma H_2_S levels of humans in the morning is predicted to be higher than in the evening. 

## 3. Materials and Methods

### 3.1. Materials

Sulfane sulfur probe 4 (SSP4) was a kind gift from Dr. Ming Xian at Washington State University, USA. Cell counting kit-8 (WST-8), sodium sulfide, and Diethylenetriamine-*N*,*N*,*N*′,*N*′′,*N*′′-pentaacetic acid (DTPA) were purchased from DOJINDO chemical laboratory, Japan. Human serum albumin and *N*,*N*-dimethyl-*p*-phenylenediamine (DPDA) were obtained from Sigma Aldrich, St. Louis, MO, USA. Zinc acetate, 2,2′-azobis(2-amidinopropane), dihydrochloride (AAPH), ascorbic acid, dimethyl sulfoxide (DMSO), hydrochloric acid (HCl), potassium hydroxide (KOH), iron (III), chloride 5,5-dithiobis-2-nitrobenzoic acid (DTNB), glutathione, and linoleic acid were purchased from FUJIFILM Wako Pure Chemical Corporation, Osaka, Japan. OneTouch^®^ was purchased from LifeScan Japan, Tokyo, Japan. Salivette^®^ was obtained from SARSTEDT, Nümbrecht, Germany. All water in assays was used deionized and distilled one. The ultraviolet plate was purchased from zell-kontakt GmbH, Nörten-Hardenberg, Germany. Hexadecyltrimethylammonium bromide (CTAB) was obtained from Tokyo Chemical Industry, Tokyo, Japan. The salivary amylase monitor and its chips were gifted by Nipro, Osaka, Japan.

### 3.2. Sample Collection

Plasma was collected by pricking fingertips (second to fourth finger) using OneTouch^®^. Plasma, saliva, and tear fluid samples were collected in the morning of the sampling day, between 8:30 am to 11:30 am. Saliva was obtained with a cotton swab placed on the hypoglottis for 1 min after brushing teeth without using toothpaste for 3 min. The cotton was then placed into a Salivette^®^ tube and centrifuged at 2000× *g* for 5 min. Saliva at the bottom of the tube was collected and used for experiments. Nasal discharge was blown into Kimwipes^®^ and centrifuged at 2000× *g* for 5 min in an empty Salivette^®^ tube. Semen was collected to 50 mL of a Falcon^®^ tube and incubated at room temperature until Liquefaction (about 30 min to 1 h). One percent of antimicrobial agent was mixed into seminal fluids. Studies involving human fluid collection were approved by the Ethics Review Committee for Human Experimentation of our institution (Tokushima University, TU, Tokushima, Japan), and informed consent was obtained from all subjects (TU-No. 3351).

### 3.3. Measuring Polysulfide by EMSP

3× EMSP solution was made by mixing ascorbic acid (792.54 mg) with 1.5 mL of water and 5 N KOH (3 mL). Samples were diluted in water and 3× EMSP was added. Then, samples were incubated at 37 °C for 4 h. After the reaction, they were mixed with 600 μL of 1% sodium acetate and centrifuged at 2300× *g* for 5 min to recover the released sulfide as a precipitate. Supernatants were removed gently and washed by 1 mL of water 3 times to remove completely peptides and proteins contained in the supernatant. After the last round of supernatant removal, water (500 μL) was added and vortexed. Protein contamination was checked using a protein determination assay. Twenty millimolar of DPDA (50 μL) in 1.2 N HCl and 30 mM of FeCl_3_ (50 μL) in 7.2 N HCl were mixed into the solution and vortexed well. The samples were centrifuged at 2300× *g* for 5 min and 200 μL of each solution were transferred into 96 well plates and absorbance was measured at 665 nm. A standard curve was constructed by using Na_2_S (15.6 to 250 μM).

### 3.4. Measuring Activities of Sperm in Semen

Twenty microliters of semen were diluted in 160 μL of 67 mM sodium phosphate buffer (pH 8.0) and mixed with 20 μL of WST-8. After the 1 h incubation at 37 °C, samples were centrifuged at 10,000× *g* for 5 min. Absorbance at 450 nm was measured on a 96 well plate.

### 3.5. Determination of Thiol Contents in Plasma

Twenty microliters of plasma were mixed into 100 μL of 5 mM DTNB in 100 mM of potassium phosphate buffer/1 mM DTPA (pH 7.0). After incubation for 60 min at room temperature, absorbance at 412 nm was measured by a plate reader (BioTek, Winooski, VT, USA). GSH (31.3 to 1000 μM) was used for constructing a standard curve.

### 3.6. Measuring Anti-Oxidative Activity Against AAPH Radical 

Anti-oxidative activity was analyzed by AAPH radical method as previously described [39]. Sixteen millimolars of linoleic acid solution was prepared by mixing 5 mL of borate buffer (50 mM, pH 9.0), 250 μL of linoleic acid, 1 mL of sodium hydroxide, and 250 mL of tween20 and diluting in a measuring cylinder to 50 mL by borate buffer (50 mM, pH 9.0). AAPH was dissolved in cold water on ice. Nine hundred and twenty microliters of phosphate buffer saline (PBS) preheated at 37 °C, 20 μL of plasma was mixed, and 10 μL of linoleic acid solution (16 mM) was added. Fifty microliters of AAPH solution (50 mM) was added and incubated for 1 h at 37 °C. After the reaction, the sample solution was dispensed into 96 wells of ultraviolet plate and absorbance, read at 234 nm. Radical elimination activity was calculated as follows:

Eliminated radical (%) = (Abs. at 234 nm of sample with AAPH − Abs. at 234 nm of sample without AAPH) × 100/(Abs. at 234 nm of PBS with AAPH − Abs. at 234 nm of PBS without AAPH)

### 3.7. Detection of Sulfane Sulfur by a Fluorescence Probe

Sulfane sulfur was detected by a fluorescence probe, SSP4, according to a previous report [40]. Plasma was diluted in 1 mL of 1 mM CTAB/PBS. 2 mL of 1 mM SSP4 in DMSO was added and incubated for 10 min at room temperature. Fluorescence intensity was measured at ex/em = 457 nm/514 nm.

### 3.8. Statistical Analysis

The statistical significance of collected data was evaluated using the ANOVA analysis followed by the Newman–Keuls method for more than 2 means. Differences between groups were evaluated by the Student’s t test. *p* < 0.05 was regarded as statistically significant. 

## 4. Conclusions

We succeeded in detecting the presence of polysulfide in various biological fluids, including semen and nasal discharge, for the first time. Each polysulfide level had no co-relationship among themselves except those between plasma and semen. These results suggest that polysulfide in each type of biological fluid was surrounded by different independent environments comprised of different protein compositions. Therefore, optimum fluid should be selected and analyzed for monitoring redox balance via measuring polysulfide. Furthermore, the effect of circadian rhythm on plasma polysulfide level warrants further investigation. 

## Figures and Tables

**Figure 1 molecules-24-01689-f001:**
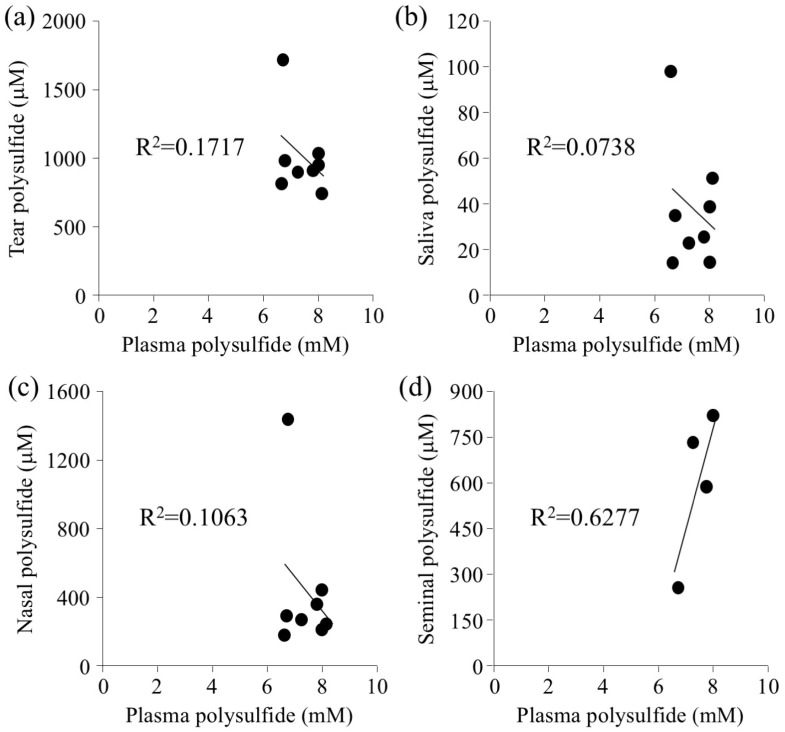
Correlation between the plasma polysulfide level and that of a biological fluid. Each polysulfide level was measured by EMSP. (**a**) tear polysulfide, (**b**) saliva polysulfide, (**c**) polysulfide in nasal discharge, and (**d**) seminal polysulfide.

**Figure 2 molecules-24-01689-f002:**
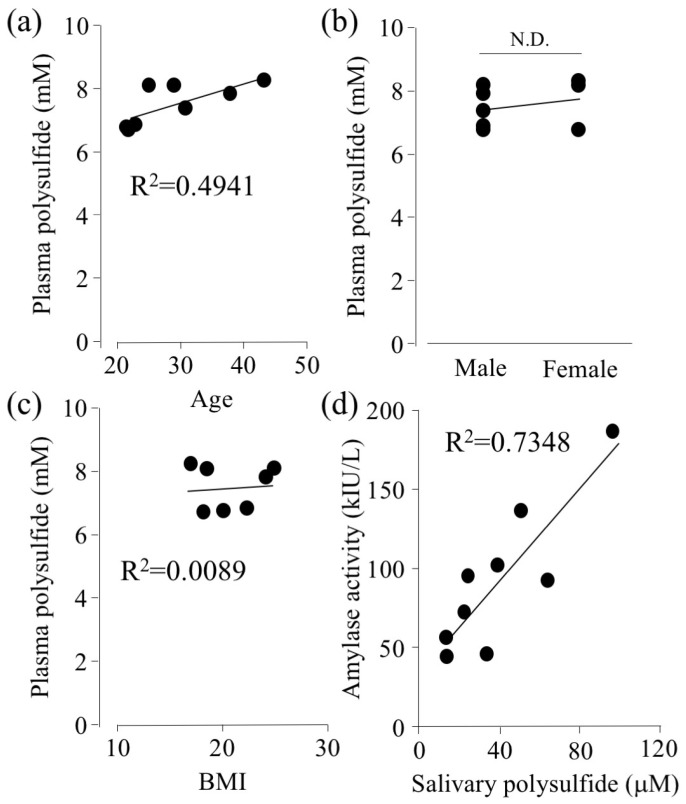
Comparison of the plasma polysulfide level to biological parameters, (**a**) age, (**b**) sex, and (**c**) BMI. (**d**) Investigation of interrelation between salivary polysulfide and amylase activity. Calculated polysulfide level was analyzed by EMSP. Amylase activity was measured by an amylase monitor.

**Figure 3 molecules-24-01689-f003:**
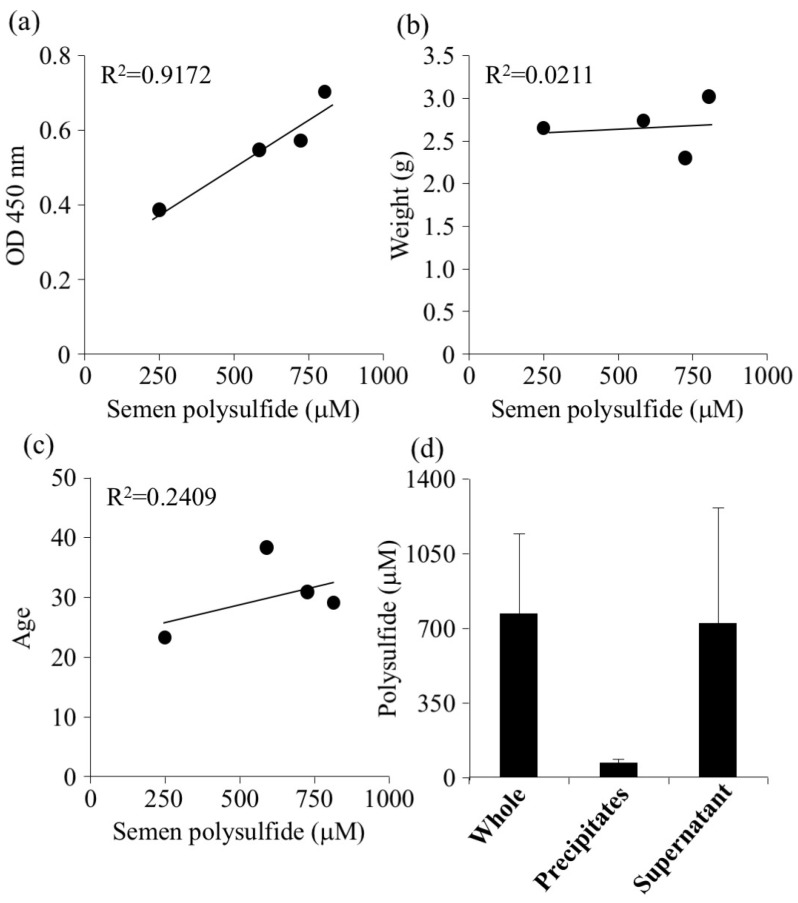
Comparing seminal polysulfide with related factors. Polysulfide levels in seminal fluids were investigated for their correlation with (**a**) sperm activity, (**b**) weight of total semen, (**c**) age. (**d**) Semen was separated by differential centrifugation at 2000× *g* for 5 min. The precipitate was washed with PBS 10 times and dissolved to the initial volume by PBS. Polysulfide level was measured by EMSP.

**Figure 4 molecules-24-01689-f004:**
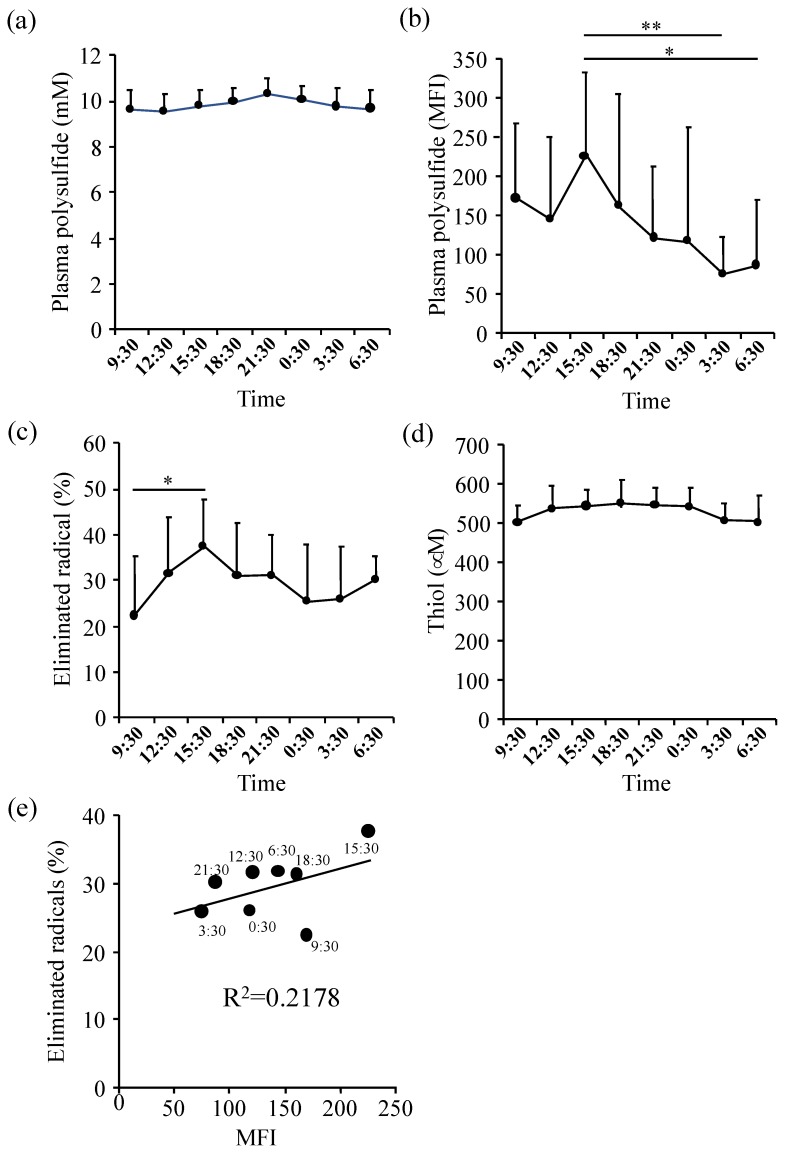
Circadian rhythms of plasma redox parameters. Diurnal variation of (**a**) plasma polysulfide measured by EMSP, (**b**) sulfane sulfur detected by SSP4, (**c**) AAPH radical elimination activity, and (**d**) thiol performed DTNB analysis was assessed. (**e**) Association between SSP4 and AAPH radical elimination activity by different time. Longitudinal axis shows the ratio of eliminated AAPH radical by plasma. Horizontal axis is the mean fluorescence intensity (MFI) of SSP4. Each point is the average score of the plasma samples collected from healthy subjects at the same time. * *p* < 0.05, ** *p* < 0.01 between groups.

**Table 1 molecules-24-01689-t001:** Characteristics of healthy human volunteers for analysis according to age, BMI, sex, and results of polysulfide levels.

n (male)	9 (5)
Age (years)	28.44 ± 7.62
BMI	20.85 ± 2.86
**Polysulfides (μM)**	
Plasma	7469.4 ± 656.68
Tear	953.55 ± 244.98
Saliva	40.854 ± 27.348
Nasal discharge	397.61 ± 399.84
Semen (*n* = 4)	594.68 ± 244.98

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
