# Peer review of "Distribution of Polysulfide in Human Biological Fluids and Their Association with Amylase and Sperm Activities"

_molecules, 2019, doi:10.3390/molecules24091689_

Round 1

Reviewer 1 Report

I have several critical remarks:

Quality of the manuscript presentation is not high.

Line 113.  “standard curve was constructed by using Na2S”.

Why you have not used polysulfides for the standard curve, as e.g. diallyl disulfide or diallyl trisulfide? The EMSP is new method and it is not tested by other researchers yet. It would improve confidence and quality of the paper if you compare some of the results obtained by EMSP method with other ones used for measurement concentration of polysulfides.     

Line 126:  the abbreviation of BMI is not explained. 

Line 127-129:  “The polysulfide level was about 7.5 mM for plasma, about 1 mM for tears, about 41 µM for saliva, about 400 µM for nasal discharge and about 600 µM for seminal fluid.” Please, mention in the text that the polysulfide level for plasma, tears and saliva were already reported in your paper: Ikeda et al. Anal Chim Acta 2017, 969, 18-25.

There are errors in figures, concentration marks should be (µM), not (ꝏM).

Please explain in details: What does it mean polysulfide concentration shown in figures? Is it concentration of RSSR or RSSSR or…?   

Line 172:  “Fig. 2a showed that plasma polysulfide was higher with increasing age within the range (22-43 years old).” Correlation is weak and probably not statistically different. This should be correctly discus.

Figure 4. Please write on y-axis that the numbers are times. In discussion, explain which results were statistically different.

Figure 4b. What is MFI?

Line 216: “The plasma polysulfide level measured by EMSP increased slightly from 12:30

pm to 21:30 and tended to decrease again by noon (Fig. 4a).” Are these results statistically different in time?

Line 218: “ In addition, we also measured polysulfide level in plasma using SSP4 which is a fluorescent probe of polysulfide.” Why you measured polysulfides by SSP4 and not by EMSP method in this case?

The method of using SSP4 and AAPH are not described.

Conclusion obtained from Fig. 4, should be corrected, since there is probably no time statistical significance in Figs. 4a,d,e.

Statistical significance is weak point of the manuscript. 

Line 360: “The correlation of each parameter is shown in Fig.6.” Where is figure 6? Is it Fig. 4e?

Author Response

Reviewer 1

1) Line 113.  “standard curve was constructed by using Na2S”.

Why you have not used polysulfides for the standard curve, as e.g. diallyl disulfide or diallyl trisulfide? The EMSP is new method and it is not tested by other researchers yet. It would improve confidence and quality of the paper if you compare some of the results obtained by EMSP method with other ones used for measurement concentration of polysulfides.    

Reply: We sincerely appreciate your insightful question. We previously clarified that 1 mol of polysulfide was detected from 1 mol of diallyl trisulfide, and that other disulfide compounds including dimethyl disulfide and cystine did not release sulfide by this method (Ikeda et al. Anal Chim Acta 2017, 969, 18-25). We tried to adopt diallyl trisulfide for standard curve several times, but the accuracy of standard curve using diallyl trisulfide solution seemed to be not high because of its volatility or organic solvent. On the other hand, Na2S is more stable especially in alkali solvent of EMSP solution. Therefore, we used Na2S for constructing the standard curve of EMSP method.

2) Line 126:  the abbreviation of BMI is not explained.

              Reply: We agree with you and have incorporated ‘body mass index (BMI)’ in the revised manuscript (Page 4, line 153).

3) Line 127-129:  “The polysulfide level was about 7.5 mM for plasma, about 1 mM for tears, about 41 µM for saliva, about 400 µM for nasal discharge and about 600 µM for seminal fluid.” Please, mention in the text that the polysulfide level for plasma, tears and saliva were already reported in your paper: Ikeda et al. Anal Chim Acta 2017, 969, 18-25.

Reply:  Thank you for your precious suggestion. We have reflected it in the revised manuscript (Page 4, line 154-157).

4) There are errors in figures, concentration marks should be (µM), not (M).

              Reply: Thank you for noting this. We are very sorry about our mistake. We have incorporated this suggestion in figures of revised manuscript.

5) Please explain in details: What does it mean polysulfide concentration shown in figures? Is it concentration of RSSR or RSSSR or…?  

              Reply: You have raised an important question. “Polysulfide concentration” means the concentration of sulfane sulfur on the polysulfide bond (RSSSR). For example, polysulfide concentration of Cys-SSS-Cys (1 mol/L) is calculated as 1 mol/L, that of Cys-SSSS-Cys (1 mol/L) is calculated as 2 mol/L.

6) Line 172:  “Fig. 2a showed that plasma polysulfide was higher with increasing age within the range (22-43 years old).” Correlation is weak and probably not statistically different. This should be correctly discus.

              Reply: Thank you for your constructive comment. We performed the statistical analysis and the p value was 0.0518. We have added ‘There is no statically significant difference (p=0.052), however, Fig. 2a showed that plasma polysulfide was tend to be higher with increasing age within the range investigated (22-43 years old)’ in revised manuscript (Page 6, line 200-202).

7) Figure 4. Please write on y-axis that the numbers are times. In discussion, explain which results were statistically different.

              Reply: Thank you for your suggestion. We have added the title of y-axis on figure 4 and the information of statistical values in revised manuscript (Page 8, line 245-253).

8) Figure 4b. What is MFI?

              Reply: Thank you for your suggestion. It was the abbreviation of “Mean fluorescence intensity”.  We have added it in revised manuscript (Page 8, line 248 and Page 9, line 259).

9) Line 216: “The plasma polysulfide level measured by EMSP increased slightly from 12:30pm to 21:30 and tended to decrease again by noon (Fig. 4a).” Are these results statistically different in time?

              Reply: We appreciate your important question. The p values between 12:30 and 21:30 were P>0.05 calculated by multiple-comparison test. Therefore, we have added the statistical results in revised manuscript (Page 8, line 245-247).

10) Line 218: “ In addition, we also measured polysulfide level in plasma using SSP4 which is a fluorescent probe of polysulfide.” Why you measured polysulfides by SSP4 and not by EMSP method in this case?

Reply: We apologize for confusing you. Figure 4a shows the data from EMSP methods, and Figure 4b shows polysulfide level of same samples measured by SSP4 method.

11) The method of using SSP4 and AAPH are not described.

              Reply: Thank you very much for your suggestion. We had written the method of SSP4 and AAPH as the supplemental data. Now we have moved them to the revised manuscript, respectively (Page 3, line 120-Page 4, line 144 and new references No. 17 and 18).

12) Conclusion obtained from Fig. 4, should be corrected, since there is probably no time statistical significance in Figs. 4a,d,e. Statistical significance is weak point of the manuscript.

              Reply: Fig. 4a and 4d showed serum polysulfide and thiols values were not time statistical significance. In contrast, Fig. 4b and 4c showed that SSP4-detectable polysulfide and radical elimination activity were time statistical significance. Interestingly, the circadian rhythm of SSP4-detectable polysulfide was similar to that of radical elimination activity (Fig. 4e). As the reviewer suggested, although Statistical significance may be weak point of the manuscript, we believe that this evidence could be important to understand that serum polysulfide possess diversity against reactive oxygen species.

13) Line 360: “The correlation of each parameter is shown in Fig.6.” Where is figure 6? Is it Fig. 4e?

              Reply: We apologize for confusing you. We corrected it (Page 10, line 261).

Reviewer 2 Report

The authors describe a valuable, yet simple quantitative detection method of polysulfides in various biological matrices. Mainstream methods use alkylation of persulfide and polysulfide cysteine derivatives followed by complicated isolation and detection of proteins. The standard method has recently been reported to alter polysulfide structure of sample proteins and result in artifact detection. This increases the importance of the process presented by the authors. The paper is well-written. Background and methods are presented in detail. Figures are correct. English language needs minor correction. A minor flaw of the paper is that it is substantially a practical application of the method described by the authors previously. Only real novelty is correlation of polysulfide data to physiological parameters. I have the following comments:

The authors did not disclose actual albumin concentration of plasma samples. Applying previous data of the authors (14.7 mol sulfide detected from 1 mol albumin) and assuming molecular weight of albumin 66500 g/mol, approximately 7.5 mM sulfide reported in the manuscript can be detected if average concentration of albumin is 33.9 g/L. The lower cutoff value of normal serum albumin is 35 g/L and samples surely contain lower than average concentrations, too. However, donors were stated to be healthy.

Methylene blue detection method involves significant acidification of samples. This release iron-bound sulfur from samples. The authors should discuss how much this could distort detected values.

It should also be discussed if peptide/protein remains in the samples undergoing spectrophotometry or is fully removed by centrifugation. Such contamination might distort results by turbidity. Usually much larger acceleration and longer time is used by centrifugation to sediment protein precipitate than shown in the manuscript (2300 g 5 min).

The authors do not disclose statistical method of regression.

Welch’s test is valid for two samples, but in case of data presented in Fig. 4 some multiple-comparison test should be used (e.g. some sort of ANOVA).

Author Response

Reviewer 2

1) The authors did not disclose actual albumin concentration of plasma samples. Applying previous data of the authors (14.7 mol sulfide detected from 1 mol albumin) and assuming molecular weight of albumin 66500 g/mol, approximately 7.5 mM sulfide reported in the manuscript can be detected if average concentration of albumin is 33.9 g/L. The lower cutoff value of normal serum albumin is 35 g/L and samples surely contain lower than average concentrations, too. However, donors were stated to be healthy.

              Reply: This is an important suggestion. Previously, we actually reported that 14.7 mol sulfide detected from 1 mol albumin, but this data was obtained from a commercial albumin (sigma). In addition, the serum albumin level in healthy volunteers were normal value. Therefore, purification step for albumin may increase the number of polysulfide in albumin. Further study regarding the important regulators on the polysulfide of albumin is needed.

2) Methylene blue detection method involves significant acidification of samples. This release iron-bound sulfur from samples. The authors should discuss how much this could distort detected values.

              Reply: Exactly as the reviewer suggested, sulfur could be released form iron-bound sulfur under acidic condition like Methylene blue solution. To avoid this reaction, all peptide/proteins including iron-bound sulfur were removed before Methylene blue assay. Therefore, detected values obtained from this EMSP methods did not almost distort by iron-bound sulfur. These information were added in the revised manuscript (Page 3, line 108-111)

3) It should also be discussed if peptide/protein remains in the samples undergoing spectrophotometry or is fully removed by centrifugation. Such contamination might distort results by turbidity. Usually much larger acceleration and longer time is used by centrifugation to sediment protein precipitate than shown in the manuscript (2300 g 5 min).

              Reply: Thank you very much for your constructive comment. After EMSP reaction, the released sulfide was recovered as a precipitate (ZnS), peptide and protein derived from endogenous fluids were removed completely by washing 3 times. In fact, we have confirmed that peptides and proteins were not detect in the reaction mixtures using a protein determination assay. The information was added in the revised manuscript (Page 3, line 111)

4) The authors do not disclose statistical method of regression.

Welch’s test is valid for two samples, but in case of data presented in Fig. 4 some multiple-comparison test should be used (e.g. some sort of ANOVA).

              Reply: Thank you for your constructive comment. We evaluated using the ANOVA analysis followed by Newman–Keuls method for more than 2 means. The information of statistical analysis was added in revised manuscript (Page 4, line 141-144).

Round 2

Reviewer 1 Report

Figure 1. Even it was corrected (see below), I can see that there are still not correct x-axis  marks. It  should be (µM), not (ꝏM). 

It may be posssible that it is seen in a such way on my computer only.

You corrected it, but I can still see it as before.   

4) There are errors in figures, concentration marks should be (µM), not (ꝏM).

              Reply: Thank you for noting this. We are very sorry about our mistake. We have incorporated this suggestion in figures of revised manuscript

Author Response

Reviewer 1

1) Figure 1. Even it was corrected (see below), I can see that there are still not correct x-axis  marks. It  should be (µM), not (ꝏM). It may be posssible that it is seen in a such way on my computer only.

    Reply: We corrected it again, but it may be a problem on the PC system. We will check again firmly in the final version. Thank you very much for your advice.